# Novel Non-Cyclooxygenase Inhibitory Derivative of Sulindac Inhibits Breast Cancer Cell Growth In Vitro and Reduces Mammary Tumorigenesis in Rats

**DOI:** 10.3390/cancers15030646

**Published:** 2023-01-20

**Authors:** Heather N. Tinsley, Bini Mathew, Xi Chen, Yulia Y. Maxuitenko, Nan Li, Whitney M. Lowe, Jason D. Whitt, Wei Zhang, Bernard D. Gary, Adam B. Keeton, William E. Grizzle, Clinton J. Grubbs, Robert C. Reynolds, Gary A. Piazza

**Affiliations:** 1Department of Biology Chemistry, Mathematics, and Computer Science, University of Montevallo, Montevallo, AL 35115, USA; 2Drug Discovery Division, Southern Research, Birmingham, AL 35205, USA; 3Department of Drug Discovery and Development, Harrison College of Pharmacy, Auburn University, Auburn, AL 36849, USA; 4Department of Biochemistry and Molecular Genetics, University of Alabama at Birmingham, Birmingham, AL 35205, USA; 5Department of Surgery, University of Alabama at Birmingham, Birmingham, AL 35205, USA; 6Department of Pathology, University of Alabama at Birmingham, Birmingham, AL 35205, USA; 7O’Neal Comprehensive Cancer Center, University of Alabama at Birmingham, Birmingham, AL 35205, USA

**Keywords:** nonsteroidal anti-inflammatory drug (NSAID), cyclooxygenase (COX), cyclic guanosine monophosphate (cGMP), phosphodiesterase (PDE), breast cancer, sulindac

## Abstract

**Simple Summary:**

This study describes a new compound, sulindac sulfide amide (SSA), which is a derivative of the nonsteroidal anti-inflammatory drug (NSAID) sulindac. NSAIDs like sulindac are effective at preventing cancer development and progression, but they are associated with dangerous side effects. SSA was rationally designed to remove the anti-inflammatory activity of its parent compound, sulindac, thereby eliminating the gastrointestinal, renal, and cardiovascular side effects associated with long-term NSAID use. Despite these changes, SSA was more potent in inhibiting growth and inducing apoptosis of breast cancer cells. SSA also inhibited mammary cancer development in rats without discernable side effects. The anti-cancer activity of SSA was associated with the inhibition of cyclic guanosine monophosphate phosphodiesterase (cGMP PDE) enzymes.

**Abstract:**

The nonsteroidal anti-inflammatory drug (NSAID) sulindac demonstrates attractive anticancer activity, but the toxicity resulting from cyclooxygenase (COX) inhibition and the suppression of physiologically important prostaglandins precludes its long-term, high dose use in the clinic for cancer prevention or treatment. While inflammation is a known tumorigenic driver, evidence suggests that sulindac’s antineoplastic activity is partially or fully independent of its COX inhibitory activity. One COX-independent target proposed for sulindac is cyclic guanosine monophosphate phosphodiesterase (cGMP PDE) isozymes. Sulindac metabolites, i.e., sulfide and sulfone, inhibit cGMP PDE enzymatic activity at concentrations comparable with those associated with cancer cell growth inhibitory activity. Additionally, the cGMP PDE isozymes PDE5 and PDE10 are overexpressed during the early stages of carcinogenesis and appear essential for cancer cell proliferation and survival based on gene silencing experiments. Here, we describe a novel amide derivative of sulindac, sulindac sulfide amide (SSA), which was rationally designed to eliminate COX-inhibitory activity while enhancing cGMP PDE inhibitory activity. SSA was 68-fold and 10-fold less potent than sulindac sulfide (SS) in inhibiting COX-1 and COX-2, respectively, but 10-fold more potent in inhibiting growth and inducing apoptosis in breast cancer cells. The pro-apoptotic activity of SSA was associated with inhibition of cGMP PDE activity, elevation of intracellular cGMP levels, and activation of cGMP-dependent protein kinase (PKG) signaling, as well as the inhibition of β-catenin/Tcf transcriptional activity. SSA displayed promising in vivo anticancer activity, resulting in a 57% reduction in the incidence and a 62% reduction in the multiplicity of tumors in the N-methyl-N-nitrosourea (MNU)-induced model of breast carcinogenesis. These findings provide strong evidence for cGMP/PKG signaling as a target for breast cancer prevention or treatment and the COX-independent anticancer properties of sulindac. Furthermore, this study validates the approach of optimizing off-target effects by reducing the COX-inhibitory activity of sulindac for future targeted drug discovery efforts to enhance both safety and efficacy.

## 1. Introduction

Accounting for 30% of newly diagnosed cancer cases and 15% of cancer-related deaths, breast cancer remains the most common cancer diagnosis and a leading cause of mortality in US women [1,2]. Incidence rates for breast cancer have remained relatively stable since 2000, with a woman’s lifetime risk of developing the disease hovering around 12.5% [1]. Improved therapeutics and treatment strategies have led to a steady decline in breast cancer related mortality since 1990, yet mortality rates remain disproportionately high in certain populations, most notably in non-Hispanic black women [1,3].

Chemoprevention is a promising approach for lowering breast cancer incidence rates, although a high degree of safety is required for any chronically administered drug. The selective estrogen receptor modulators (SERMs) tamoxifen and raloxifene are FDA approved for the prevention of estrogen receptor positive cancers in high-risk populations. Although not currently approved for this indication, strong evidence also supports the use of the aromatase inhibitors (AIs) exemestane and anastrazole for breast cancer prevention [4,5,6]. In 2013, the American Society of Clinical Oncology revised their clinical practice guidelines to encourage oncologists to discuss chemopreventive drugs with high risk patients [7]; this guideline was mirrored by the US Preventive Services Task Force in 2019 [8].

Unfortunately, SERMs and AIs are only effective for preventing estrogen receptor positive cancers and are associated with poor compliance and significant adverse effects, including thromboembolic events and endometrial cancer [4,5,6,7,8]. Only an estimated 15% of US women are candidates for chemoprevention, and just 5% have a favorable risk–benefit profile [4]. This limited utility significantly hampers the ability of SERMs and AIs to broadly impact breast cancer incidence, demonstrating the need to identify new targets and drug candidates for breast cancer chemoprevention.

Nonsteroidal anti-inflammatory drugs (NSAIDs) have long been studied for their ability to reduce the risk of multiple types of cancer [9]. While their effects on breast cancer appear to be low and nuanced, several studies have shown that long-term, high dose use of certain NSAIDs is associated with a significant reduction in breast cancer risk [10,11,12,13]. Unlike SERMs and AIs, the chemopreventive effects of NSAIDs do not depend on hormone receptor status [12], suggesting that these drugs may have a broader chemopreventive utility. However, like SERMs and AIs, the long-term, high dose use of NSAIDs is associated with potentially life threatening adverse effects including gastrointestinal perforation and thromboembolic events due to their cyclooxygenase (COX) inhibitory activity [14]. However, the mechanism responsible for the antineoplastic activity of NSAIDs may be at least partially independent of COX inhibition, suggesting the potential to uncouple these mechanisms and produce novel chemopreventive agents with improved efficacy and reduced toxicity [15,16,17,18,19].

The cyclic guanosine monophosphate (cGMP) signaling pathway has been identified as one such potential COX-independent NSAID target. Both the COX-inhibitory sulfide and non-COX inhibitory sulfone metabolites of the non-selective COX-1 and COX-2 inhibitor sulindac inhibit cGMP phosphodiesterases (PDEs), the enzymes responsible for degrading the second messenger cGMP, at concentrations that also inhibit cancer cell growth [15,20,21,22,23,24,25]. As a result, intracellular cGMP levels increase, activating cGMP-dependent protein kinase (PKG) and inducing cell cycle arrest and apoptosis [15,16,20,21,22,23,24,25,26]. Because certain cGMP PDE isozymes such as PDE5 and PDE10 have been reported to be overexpressed in cells during the early stages of tumorigenesis, the effects of sulindac on cGMP/PKG signaling appears to be highly selective for inducing apoptosis of neoplastic cells while having minimal effects on normal cells and tissue [20,21,22,27,28].

Through an extensive medicinal chemistry screening effort, we have generated a chemically diverse series of novel sulindac derivatives that share an indene scaffold [29,30]. One derivative, a N, N-dimethylethyl amine referred to as sulindac sulfide amide (SSA), has been evaluated in models of colon, prostate, and lung cancer, where it was shown to have negligible COX-inhibitory activity but enhanced antineoplastic activity [31,32,33]. Here, we describe the antineoplastic activity of SSA using in vitro and in vivo models of breast cancer. Although SSA has reduced potency to inhibit COX enzymes, it more potently inhibited growth and induced apoptosis of human breast cancer cell lines and prevented tumor formation in the N-methyl-N-nitrosourea (MNU) rat model of breast tumorigenesis. Mechanistic studies demonstrated that the anticancer activity of SSA was strongly associated with the activation of cGMP/PKG signaling. PDE5, a potential PDE isozyme target, was highly expressed in both human breast cancers and MNU-induced rat mammary tumors. Interestingly, the subcellular localization of PDE5 varied between normal breast tissue, with predominantly cytoplasmic localization, and breast tumors, where the enzyme is localized in the nucleus.

## 2. Materials and Methods

### 2.1. Experimental Design and Data Analysis

SSA was synthesized as described previously [30]. All biochemical and cell-based experiments were performed with a minimum of three replicates per data point and repeated a minimum of three times. Graphs were constructed from a representative experiment using GraphPad Prism 5 software. Error bars represent standard error of the mean (SEM). The calculation of *p* values was done by comparing the specified treatment group to vehicle treated control using a student’s *t* test for cell-based assays and two-way ANOVA for in vivo study. Drug effects on cell growth, apoptosis, and enzyme activity were measured and the potency expressed as an IC_50_ value, which is the concentration resulting in 50% inhibition when compared to the vehicle control as calculated by GraphPad Prism 5 using a four-parameter logistic equation.

### 2.2. Molecular Modeling

A model of PDE5 constructed from the crystal structure of tadalafil bound to the catalytic domain of PDE5 was obtained from the protein databank (PDB ID: 1UDT). Schrödinger Suite 2008 was used for molecular modeling studies using the induced fit protocol and parameters described in [31,34]. The docking protocol and parameters were first validated by docking tadalafil to the PDE5 catalytic domain; they were then used to model SSA docking in the catalytic domain of PDE5. 

### 2.3. Biochemical Assays

COX-1 and COX-2 activities were measured using purified ovine COX-1 and COX-2 with colorimetric assay kits obtained from Cayman Chemical Company (Ann Arbor, MI, USA) according to the manufacturer’s protocol. After the addition of arachidonic acid and incubation at 25 °C for 5 min, COX activity was determined by measuring absorbance at 590 nm.

PDE activity was measured using the Molecular Devices (San Diego, CA, USA) IMAP fluorescence polarization assay as previously described [15]. Recombinant PDE5 was purchased from BPS Biosciences (San Diego, CA, USA).

### 2.4. Cell-Based Assays

The human breast tumor cell lines Hs578t, MCF7, MDA-MB-231, SKBr3, and ZR75-1 were obtained from ATCC (Manassas, VA, USA) and grown in RPMI 1640 media containing L-glutamine, 5% fetal bovine serum, and penicillin/streptomycin at 37 °C in a humidified atmosphere with 5% CO_2_.

Growth inhibition was measured using the Promega (Madison, WI, USA) Cell Titer Glo Assay following manufacturer’s specifications. First, 96-well tissue culture treated microtiter plates were seeded at a density of 5000 cells per well and incubated at 37 °C for 24 h prior to drug treatment. After treatment, plates were incubated 72 h prior to assay.

Caspase activation was measured using Promega’s Caspase 3/7 Glo Assay following the manufacturer’s specifications. First, 96-well tissue culture treated microtiter plates were seeded at a density of 10,000 cells per well and incubated at 37 °C for 24 h prior to drug treatment. After treatment, plates were incubated 6 h prior to assay.

A terminal deoxynucleotidyl transferase dUTP nick end labeling (TUNEL) assay was performed in SKBr3 cells using the Invitrogen APO-BrdU TUNEL Assay (Waltham, MA, USA) according to the manufacturer’s protocol, and fluorescence was measured using a Guava EasyCyte (Hayward, CA, USA) flow cytometry system. First, 10 cm tissue culture treated dishes were seeded at a density of 2,000,000 cells per dish and incubated at 3 °C for 24 h prior to drug treatment. After treatment, cells were incubated for 24 h and then collected and fixed with 4% paraformaldehyde for 15 min on ice, washed with phosphate buffered saline (PBS), and incubated in 70% ethanol at −20 °C for 4–24 h prior to assay.

Viable mini-tumors (1 mm in size) of MDA-MB-231 breast tumor cells were generated using the three-dimensional human Biogel culture system from Vivo Biosciences (Birmingham, AL, USA). Accordingly, 20,000 cells were mixed with HuBiogel beads and cultivated in rotary bioreactors for 3 days. Live cell imaging was performed using Calcein- acetoxymethyl (calcein-AM) staining.

Intracellular levels of cGMP were measured in MDA-MB-231 cells after 30 min of drug treatment, as previously described [15].

Tcf transcriptional activity was measured in Hs578t cells. Cells were seeded in 24-well tissue culture treated plates. After 24 h of incubation, cells were transiently transfected with 0.1 µg of Promega β-galactosidase-expressing vector and 0.1 µg of Super8XTOPFlash construct, which was kindly provided by Randall T. Moon from the University of Washington (Seattle, WA, USA). After 24 h of transfection, cells were treated with compound and incubated for an additional 24 h. Cells were lysed, and luciferase and β-galactosidase activities were measured using assay systems from Promega (Madison, WI, USA), following the manufacturer’s instructions. Luciferase activity was normalized to β-galactosidase activity.

### 2.5. Western Blotting

All antibodies were obtained from Cell Signaling Technologies (Danvers, MA, USA). Cells were harvested and vortexed in ice cold lysis buffer containing 20 mM tris acetate, 5mM magnesium acetate, 1 mM ethylene glycol-bis(β-aminoethyl ether)-N,N,N′,N′-tetraacetic acid (EGTA), 0.8% Triton X-100, 50 mM sodium fluoride, 1.25 mM sodium vanadate, and protease inhibitor cocktail at pH 7.4. Lysates were centrifuged at 10,000× *g* for 10 min at 4 °C. Protein contents were determined using the Thermo/Pierce (Waltham, MA, USA) bicinchoninic acid (BCA) protein assay following the manufacturer’s specifications.

Proteins (15 µg) from lysed cells were separated by sodium dodecyl sulfate polyacrylamide gel electrophoresis (SDS-PAGE) in a 12% polyacrylamide gel followed by electrophoretic transfer to a nitrocellulose membrane. The membranes were blocked with 5% nonfat dry milk in tris-buffered saline (TBS) containing 0.05% Tween-20. Membranes were incubated with primary antibody at 4 °C overnight followed by incubation with secondary antibody conjugated to horseradish peroxidase for 2 h at room temperature. Protein bands were visualized using Millipore Immobilon enhanced chemiluminescence (ECL) substrate on HyBlot CL autoradiography film. 

### 2.6. In Vivo Efficacy

In vivo efficacy was evaluated using the N-methyl-N-nitrosourea (MNU) model of mammary carcinogenesis, as previously described [35]. Briefly, female Sprague-Dawley rats were obtained from Envigo (formerly Harlan Sprague-Dawley, Inc.) (Indianapolis, IN, USA) at 28 days of age and placed on a Teklad 4% fat diet on the day of arrival. Within one week of arrival, rats were randomly assigned to groups (15 rats/group), ear marked for identification, and weighed to ensure statistically similar average body weights per group. At 50 days of age, rats received one IV injection of MNU (75 mg/kg) via the jugular vein. Rats were administered 800 or 1200 ppm of SSA or vehicle control in diet beginning 3 days after MNU administration. Rats were weighed once per week, palpated for mammary tumors twice per week, and checked daily for signs of toxicity. The study was terminated at 126 days after MNU administration. All mammary tumors were weighed when removed at necropsy, weighed, and processed for histological classification and analysis. The protocol used for this study was approved by the University of Alabama at Birmingham (UAB) IACUC where the study was performed.

### 2.7. Immunohistochemistry

Rat mammary tumor tissues were obtained from Sprague-Dawley rats at 126 days after MNU administration. Noncancerous tissues were obtained from age-matched rats that were not exposed to MNU. None of the rats was treated with additional drugs or chemicals. Human tissues were obtained from the Cooperative Human Tissue Network at UAB. The use of all tissues was approved by IACUC and IRB at UAB. All tissues were formalin fixed and paraffin embedded. Tissues were de-paraffinized in xylene and rehydrated in graded alcohols. For antigen retrieval, the slides were pressure cooked for 10 min. Slides were incubated 1 h with primary antibody. The biotinylated secondary antibody and the avidin/horseradish peroxidase label were each applied for 10 min (BioGenex Laboratories, Inc., Freemont, CA, USA). The antigen-antibody reaction was visualized after diaminobenzidine had been applied for 7 min. The slides were counterstained with hematoxylin for 1 min. Positive controls were included in each experiment; negative controls were obtained by omitting the primary antibody. Slides were then dehydrated in alcohols and cleared in three xylene baths before being mounted with Permount media.

## 3. Results

### 3.1. SSA Lacks COX-Inhibitory Activity but Inhibits Breast Tumor Cell Growth In Vitro

The NSAID sulindac is an orally bioavailable prodrug that is metabolized by liver enzymes to the inactive sulfone metabolite and the active, COX-inhibitory sulfide metabolite [27]. As shown in Figure 1A, SSA is a N,N-dimethylethyl amide derivative of sulindac sulfide (SS). While SS inhibited COX-1 with an IC_50_ value of 1.2 µM and COX-2 with an IC_50_ value of 9.0 µM, the concentrations of SSA required to inhibit COX-1 or COX-2 were appreciably higher, with IC_50_ values of 81.6 µM and greater than 200 µM, respectively (Figure 1B).

Despite significantly reduced COX-inhibitory activity, SSA was appreciably more potent than SS in inhibiting growth and inducing apoptosis of human breast cancer cells. As shown previously, SS exhibited IC_50_ values of 58.8–83.7 µM for human breast cancer cell growth [15]. However, using the same human breast cancer cell lines and treatment conditions, SSA displayed IC_50_ values of 3.9–7.1 µM (Figure 2A). 

The inhibition of breast cancer cell growth by SSA was attributed to the induction of apoptosis, as demonstrated through caspase activation and DNA cleavage. After just 6 h of SSA treatment, the activity of initiator caspases 3 and 7 increased by 2 to 6-fold in Hs578t, MDA-MB-231, SKBr3, and ZR75-1 cells (Figure 2B). MCF7 cells showed no increase in caspase activation, which is consistent with this cell line’s known deficiencies in caspase expression [36]. SKBr3 cells demonstrated a significant increase in DNA cleavage, as measured by TUNEL assay after 24 h of SSA treatment (Figure 2C). 

Consistent with the results of cell growth in two-dimensional (2D) culture, SSA treatment also markedly reduced MDA-MB-231 cell growth in 3D HuBiogel (Figure 2D). Although slightly higher concentrations of SSA were required to induce apoptosis and inhibit 3D growth compared with 2D growth, this difference was likely due to the shorter duration of time needed to measure the biochemical markers of apoptosis (6 h) compared with the time required to reduce the number of viable cells (72 h).

### 3.2. SSA Activates cGMP/PKG Signaling and Attenuates Wnt/β-Catenin Transcriptional Activity

Because the antineoplastic activities of SSA’s parent compound (SS) were previously reported to be associated with PDE5 inhibition and activation of cGMP signaling [15], we determined if SSA had similar effects. In molecular modeling studies, SSA docked in an energetically favorable manner to the active site of PDE5 (Figure 3A). Interestingly, while the commercially available PDE5 inhibitor tadalafil is shown to interact with a single active site residue, i.e., glutamine 817, SSA was predicted to interact with three residues, i.e., tyrosine 612, asparagine 662, and glutamate 682. This suggests that SSA has a different mode of binding and mechanism of enzyme inhibition compared with known PDE5 inhibitors, which have only modest anticancer activities [37].

Consistent with molecular modeling studies, SSA significantly inhibited the activity of recombinant PDE5 and cGMP PDE in lysates of MDA-MB-231 and SKBr3 breast cancer cells (Figure 3B). While 25 µM SSA inhibited recombinant PDE5 by 55%, 100 µM was required to inhibit cGMP hydrolysis by 38% and 65% in the MDA-MB-231 and SKBr3 whole cell lysates, respectively. SSA more potently increased intracellular cGMP levels in intact MDA-MB-231 breast cancer cells at concentrations that were comparable with growth IC_50_ values (Figure 3C). 

The cGMP-dependent protein kinase (PKG) is a major effector of cGMP signaling in human breast cancer cells [26,38,39]. To further determine if SSA inhibits cGMP PDE in cells, and to identify whether the observed increase in intracellular cGMP levels was sufficient to activate PKG, we measured phosphorylation of vasoactivator-stimulated phosphoprotein (VASP) at the PKG-specific serine 239 residue. SSA significantly increased the level of VASP phosphorylation within the same time period as it increased cGMP levels (Figure 3D). VASP phosphorylation peaked after 1 h of treatment and then steadily decreased over 7 h of subsequent treatment.

Activation of PKG has been reported to increase phosphorylation and subsequent degradation of β-catenin, an important activator of Tcf/Lef transcription factors [16]. To determine the potential involvement of these downstream effects in the mechanism of SSA, we evaluated the expression of the Tcf/Lef targets cyclin D1 and survivin (Figure 3D). Cyclin D1 protein levels decreased appreciably after just 1 h of 7.5 µM SSA treatment, whereas survivin protein levels decreased appreciably after 7 h of treatment. Involvement of Tcf/Lef transcription factors was confirmed via a reporter assay. As shown in Figure 3E, Tcf transcriptional activity was reduced by 14% with 5 µM SSA and by 60% with 20 µM SSA treatment. The effects of SSA on the transcriptional activity of Tcf/Lef and on cyclin D1 and survivin expression occurred at concentrations and time points comparable to those necessary for intracellular increases in cGMP and activation of PKG.

### 3.3. SSA Inhibits Tumor Formation in the MNU Rat Model of Mammary Carcinogenesis

We evaluated the chemopreventive efficacy of SSA in the MNU rat model of mammary carcinogenesis to determine the in vivo relevance of our in vitro findings. Plasma levels of SSA in Sprague-Dawley rats fed a diet containing 1000, 2000, or 3000 ppm of SSA were found to exceed the concentrations necessary for SSA to activate cGMP signaling and inhibit growth in breast cancer cells in vitro (Figure 4A). Dietary dosages of 800 and 1200 ppm were selected for further evaluation. 

Neither dose of SSA resulted in a significant change in body weight over the 126-day study (Figure 4B). Both 800 and 1200 ppm of SSA resulted in marked reductions in the incidence (Figure 4C) and multiplicity (Figure 4D) of MNU-induced tumors. In contrast, 93% of rats in the vehicle control group had palpable tumors, and the group averaged 1.93 tumors per rat. Treatment with 800 ppm SSA resulted in a 22% reduction in incidence and 31% reduction in multiplicity compared to the vehicle control, while treatment with 1200 ppm SSA resulted in a 43% reduction in incidence and 62% reduction in multiplicity compared to the vehicle control. Apart from the changes in incidence and multiplicity, there were no observable differences in tumor characteristics between control and treatment groups.

### 3.4. PDE5 Is Differentially Expressed in Normal and Cancerous Breast Tissues

PDE5 was previously identified as the predominant cGMP PDE isozyme in human breast cancer cells [21]. As such, we evaluated the expression of PDE5 by immunohistochemistry in sporadic human breast cancers (Figure 5A) and MNU-induced mammary tumors of Sprague-Dawley rats (Figure 5B). In all tissue samples, PDE5 was highly expressed in the epithelial and tumor cells with little expression in stromal, endothelial, and inflammatory cells. There were two consistent expression patterns observed. The first pattern, indicated by a solid black arrowhead, demonstrated diffuse staining throughout the cytosol and nucleoplasm, with slightly stronger staining in the nucleoplasm. This pattern was associated with marked perinuclear staining and discrete punctate regions within the nucleus and was predominantly displayed in human breast cancer samples. The second expression pattern, indicated by a white arrowhead, demonstrated predominantly nuclear with some cytosolic labeling. This staining pattern was mostly displayed in the non-cancerous human breast tissue samples as well as normal and tumorigenic rat mammary samples. Both human and rat tumor samples displaying this pattern had a greater proportion of nuclear to cytosolic labeling compared to the non-tumorigenic samples displaying this pattern, suggesting that differences in the localization of PDE5 may reflect differences in PDE5 function in normal cells compared to cancerous cells.

## 4. Discussion

Epidemiological studies have demonstrated that the long-term, high dose use of NSAIDs such as sulindac can lower the risk of breast cancer [9,10,11,12]. A prospective cohort study found that regular use of aspirin reduced a woman’s risk of breast cancer by 39%, an association independent of familial risk, patient age, and tumor receptor status [12]; however, the toxicities that result from COX inhibition preclude the use of NSAIDs for chronic indications. We and others have reported that the antineoplastic activity of sulindac may be attributed to one or more COX-independent mechanisms [15,16,17,18,19,20,21,26,27,28,31,32,33] and have studied chemical modifications that can block COX binding while enhancing binding for the underlying oncogenic target(s).

As previously reported [31], by replacing the negatively charged carboxyl moiety of sulindac sulfide with a dimethylaminoethyl-amide moiety, we effectively blocked the ability of the compound to inhibit either COX-1 or COX-2 while exponentially improving its potency to inhibit breast cancer growth in vitro and in vivo. This relatively simple chemical modification demonstrates that the COX-inhibitory and antineoplastic activities of sulindac can effectively be uncoupled to potentially reduce toxicity while enhancing anticancer activity. In addition, other, more complex chemical derivations have been discovered that also demonstrate improved anticancer potency despite lacking COX-inhibitory activity [23,29,34,40,41,42]. While these other derivatives have additional chemical modifications, all involve substitutions to the carboxylic acid residue, demonstrating the importance of carboxylic acid for COX inhibition and the ability to make certain amide substitutions that improve anticancer activity [31].

Numerous studies have investigated the COX-independent mechanism responsible for the anticancer activities of NSAIDs and their derivatives. While several different cellular pathways have been implicated, activation of GMP/PKG signaling and subsequent attenuation of oncogenic β-catenin-induced transcription may be one of the most important for many malignancies that are driven by aberrant Wnt signaling. cGMP PDE isozymes have been shown to be overexpressed during early stages of carcinogenesis, and Wnt signaling has a known impact on tumorigenesis, prognosis, and resistance to treatment [9]. Furthermore, activation of cGMP/PKG signaling suppresses the growth of human breast cancer cells [38,43,44,45,46,47,48,49]. SSA, like SS, was shown to inhibit cGMP PDE activity, raise intracellular levels of cGMP, activate PKG, and reduce Tcf transcriptional activity at concentrations that correlated with its ability to inhibit growth and to induce apoptosis of human breast cancer cells, adding further evidence of this COX-independent antineoplastic mechanism.

Differences that have been reported in PDE5 expression and activity in human breast cancer cells have led many to speculate that inhibition of PDE5 is an important pharmacologic mechanism for activating cGMP signaling in these cells [15,46,47,48,49]. PDE5 inhibitors have demonstrated anticancer activity, and regular use of PDE5 inhibitors is associated with reduced risk of colorectal cancer [37,49,50,51]. PDE5 inhibitors also show promise as chemoadjuvant therapies through their ability to inhibit multi-drug resistance pathways [52,53]. However, with traditional PDE5 inhibitors like sildenafil and tadalafil, the concentrations necessary to affect cancer cell growth in vitro and in vivo are substantially higher than the concentrations required to inhibit PDE5 [37]. We have speculated that this reduced potency is due to the drugs being unable to access PDE5 in breast cancer cells, perhaps as a result of decreased import or increased efflux of the drugs when compared to sensitive cells like smooth muscle cells. Alternatively, it may be due to the co-expression of other cGMP degrading PDE isozymes in cancer cells such as PDE10 [15,54].

The results from this study provide new insights into this discrepancy. First, our molecular modeling studies suggest that, while both tadalafil and SSA dock in the active site of PDE5, the interactions of these two compounds with amino acids in the protein’s active site are distinct. SSA appears to form additional interactions within the pocket. While more studies into the biochemical activities of SSA are needed, it is possible that SSA results in a more complete and longer lasting inhibition of PDE5, which may be necessary for the activation of cGMP signaling in breast cells. Second, our immunohistochemistry data revealed a unique subcellular localization pattern for PDE5 in human and rat mammary tissues. While studies of smooth muscle cells have demonstrated that PDE5 is localized in cytosolic and vesicular compartments [55], the breast tissues examined in this study showed more prominent nuclear and perinuclear localization patterns. This difference in subcellular localization of PDE5 in normal as compared to cancer cells may also reflect a unique function in cancer cells that could account for the poor ability of traditional PDE5 inhibitors to suppress cancer cell proliferation and induce apoptosis.

An important feature of NSAIDs and their non-COX-inhibitory derivatives is their ability to inhibit the growth of both estrogen receptor expressing and non-expressing cells, an activity that both SERMs and AIs lack. The panel of human breast cancer cells evaluated in this study represents a broad range of breast cancer molecular subtypes, including luminal A, luminal B, HER2-enriched, and triple negative subtypes [56,57,58,59]. Despite their molecular variability, each of the cell lines was sensitive to the growth inhibitory and apoptosis-inducing activities of SSA, and there were minimal differences in potency amongst the different molecular subtypes represented. These results suggest that NSAID derivatives may prevent a wider range of breast cancer types compared with current chemopreventive agents. Furthermore, because SSA lacks the COX-inhibitory activity of traditional NSAIDs, SSA is also likely to have a more favorable side effect profile, demonstrating SSA’s much broader chemopreventive potential when compared with SERMs and AIs.

The cancer chemopreventive activity and tolerance of SSA was further demonstrated in the MNU model of breast tumorigenesis. Both tumor incidence and multiplicity significantly decreased in a dose dependent manner, and there was no discernable toxicity observed in the rats despite the long-term treatment. Unlike the broad range of molecular subtypes represented in the in vitro experiments, this in vivo study was limited to evaluating the efficacy of SSA for the prevention of the low grade, estrogen receptor expressing mammary tumors that were induced by MNU [60,61]. Additional studies are necessary to demonstrate the in vivo chemopreventive efficacy of SSA for other tumor subtypes, particularly HER2-enriched and triple negative cancers.

## 5. Conclusions

The studies presented here provide compelling evidence that specifically and rationally modifying sulindac is a promising strategy for developing safer and more effective breast cancer chemopreventive agents. Furthermore, the unique PDE5 expression patterns observed in breast cancer tissues and the mounting evidence of the anticancer properties of cGMP signaling warrant further investigation into this signaling pathway as a potential target for future anticancer drug discovery efforts.

## Figures and Tables

**Figure 1 cancers-15-00646-f001:**
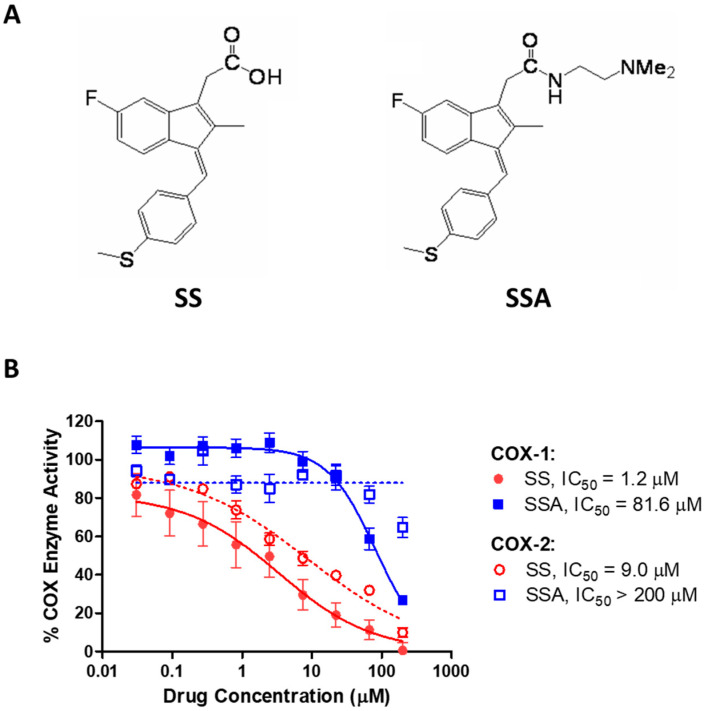
Comparison of sulindac sulfide (SS) and sulindac sulfide amide (SSA). (**A**) Structures of the parent compound SS (left) and the novel dimethylethyl amide derivative SSA (right). (**B**) Inhibitory activity of SS (●, ○) and SSA (■, □) against purified COX-1 (solid lines) and COX-2 (dashed lines) enzymes.

**Figure 2 cancers-15-00646-f002:**
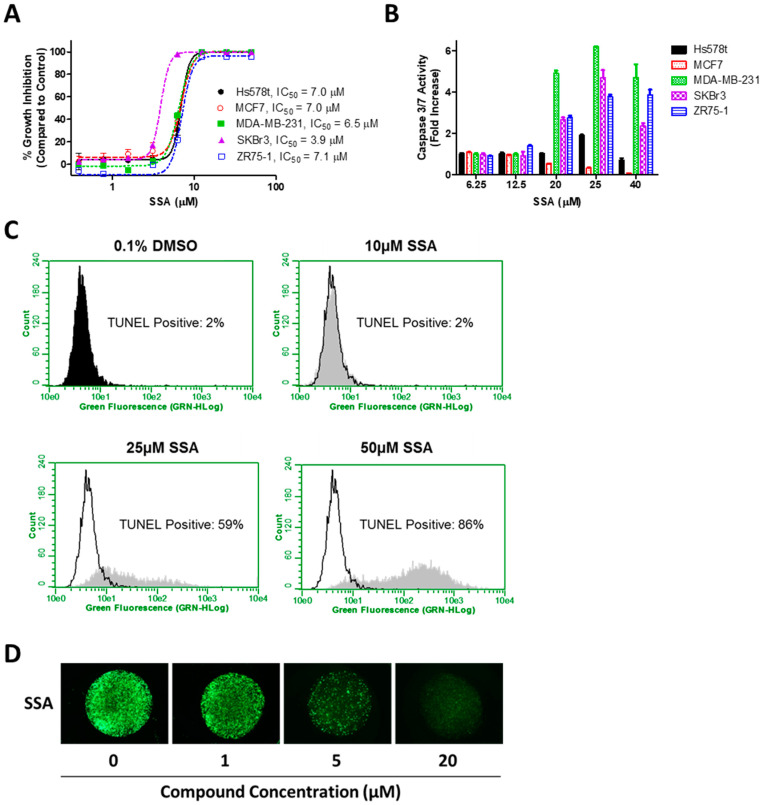
In vitro anticancer activity of SSA. (**A**) Effect of SSA on 2D growth of human breast cancer cell lines after 72 h of treatment. (**B**) Effect of SSA on the activity of initiator caspases 3 and 7 in human breast cancer cell lines after 6 h of treatment. (**C**) Effect of SSA on late apoptosis, as measured by DNA fragmentation via TUNEL assay in SKBr3 breast cancer cells after 24 h of treatment. (**D**) Live imaging via calcein AM staining of MDA-MB-231 breast cancer cells grown in 3D HuBiogel.

**Figure 3 cancers-15-00646-f003:**
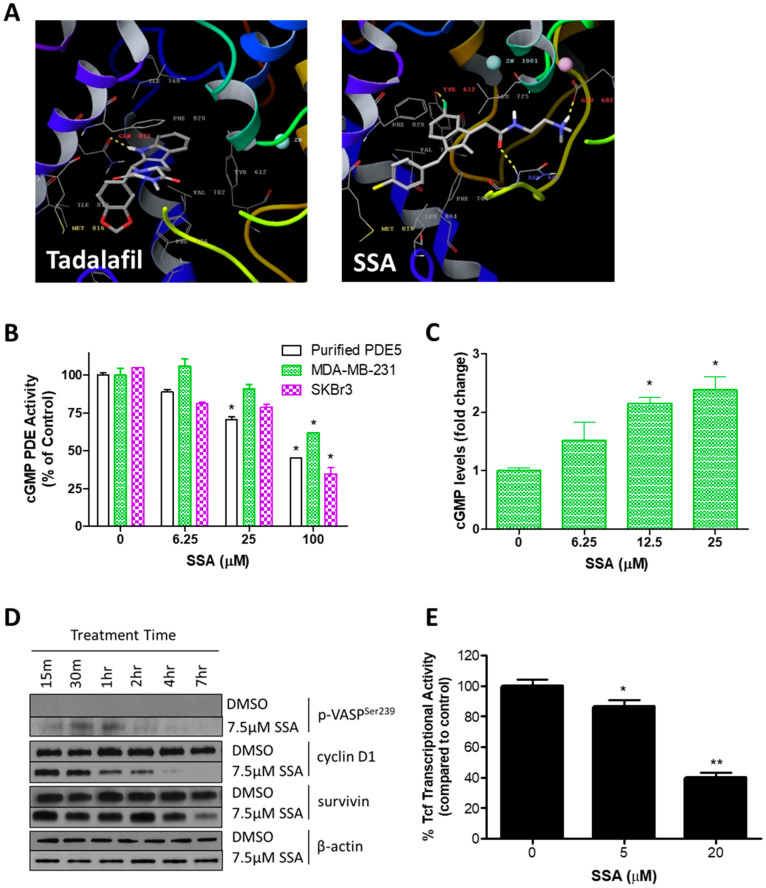
Activation of cGMP Signaling with SSA. (**A**) Molecular modeling of the conventional PDE5 inhibitor tadalafil (left) and SSA (right) in the active site of PDE5. (**B**) Effect of SSA on cGMP hydrolysis by purified PDE5 and MDA-MB-231 and SKBr3 human breast cancer cell lysates. (**C**) Effect of SSA on intracellular cGMP concentrations in MDA-MB-231 breast cancer cells after 30 min of treatment. (**D**) Immunoblotting of MDA-MB-231 breast cancer cells treated with SSA or DMSO vehicle control 15 min, 30 min, 1 h, 2 h, 4 h, and 7 h following SSA treatment. Phospho-VASP^Ser239^ was evaluated as a marker of PKG activity. β-actin was included as a loading control. Original Western blot figure can be found in Appendix A. (**E**) Effect of SSA on Tcf/Lef promoter activity in Hs578t breast cancer cells. A single asterisk (*) indicates *p* < 0.05. A double asterisk (**) indicates *p* < 0.001.

**Figure 4 cancers-15-00646-f004:**
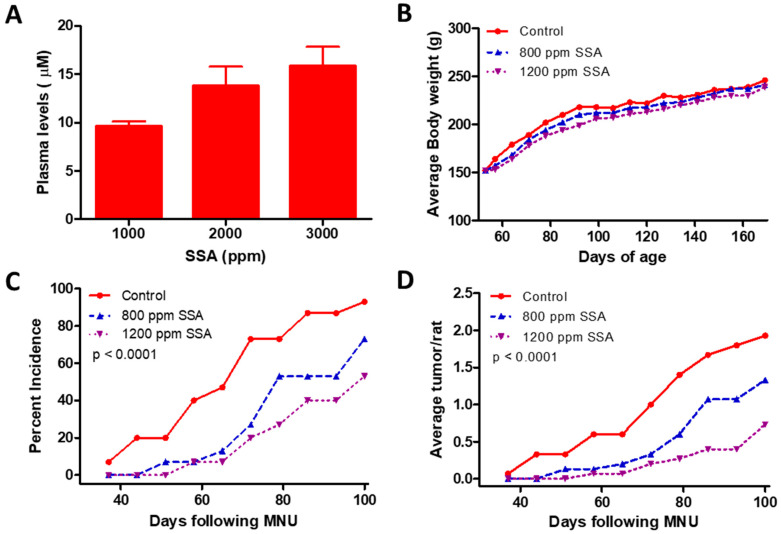
In vivo evaluation of the chemopreventive activity of SSA in MNU-induced rat model of mammary tumorigenesis. (**A**) Plasma levels of SSA achieved after feeding of SSA at various levels. (**B**) Effects of SSA feeding on body weight of Sprague-Dawley rats. (**C**) Tumor incidence and (**D**) tumor multiplicity in rats fed with vehicle control or SSA diet beginning 3 days after MNU exposure.

**Figure 5 cancers-15-00646-f005:**
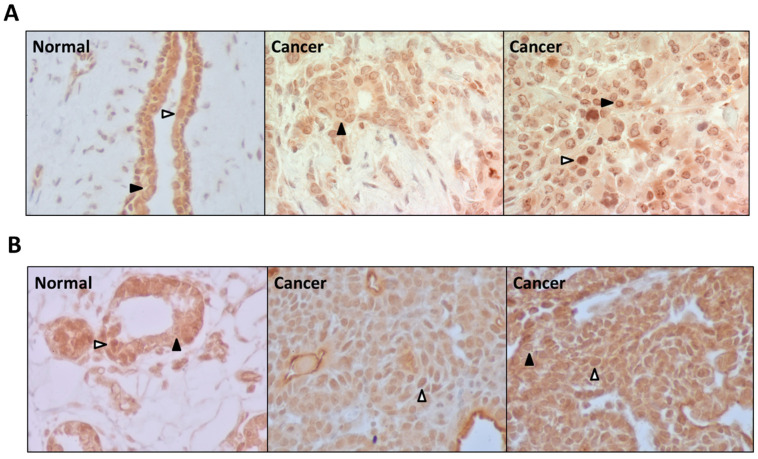
Immunohistochemical evaluation of PDE5 expression in noncancerous and cancerous breast tissue samples, 400× magnification. (**A**) Human breast tissue samples include uninvolved breast tissue from a patient diagnosed with breast cancer (left) and cancerous biopsy samples from different patients (middle, right). (**B**) Sprague-Dawley rat mammary tissue samples include noncancerous tissue from a rat not exposed to MNU (left) and cancerous tissue from two separate rats after exposure to MNU (middle, right). Black arrowheads (▲) indicate diffuse cytosolic and nuclear staining with marked perinuclear localization. White arrowheads (△) indicate predominantly nuclear staining with minimal cytosolic labeling.

## Data Availability

All data generated or analyzed during this study are included in this published article. Raw data generated during and/or analyzed during the current study are available from the corresponding author on reasonable request.

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
