# Peer review of "Novel Non-Cyclooxygenase Inhibitory Derivative of Sulindac Inhibits Breast Cancer Cell Growth In Vitro and Reduces Mammary Tumorigenesis in Rats"

_cancers, 2023, doi:10.3390/cancers15030646_

Round 1

Reviewer 1 Report

I have reviewed the manuscript entitled” Novel non-cyclooxygenase inhibitory derivative of sulindac in habits breast cancer cell growth in vitro and reduces mammary tumorigenesis in rats” by Tinsley et al. This is an interesting study; the authors have collected a unique dataset using a proper methodology. The paper is well-written and structured. So interesting results.  I hardly can find any inappropriate sections and unclear results the only part I would suggest rechecking is in Figure 4. B and D - are there any statistically significant changes in the tumor size? If so, I suggest to add it on the line graph

Author Response

Thank you for your honest review. Below is a summary of points from your comments and our responses.

Point 1: I hardly can find any inappropriate sections and unclear results the only part I would suggest rechecking is in Figure 4. B and D - are there any statistically significant changes in the tumor size? If so, I suggest to add it on the line graph

Response 1: Figure 4 was updated to include statistical data, specifically to panels C and D. The methods were also updated to include information about the statistical analysis.

Reviewer 2 Report

In the work, I identified that different mammary lineages representing the various molecular subtypes were used, however the number of satisfactory lineages in this study, which are four, I consider insufficient to cover all subtypes, so I understand that a larger number of mammary lineages should be used, as well as specifying each lineage.

 The methodology described requires more information such as the number of animals used, informing tumor characteristics (subtype) after induction of the drug, N-methyl-N-nitrosourea (MNU).

Author Response

Thank you for your honest review. Below is a summary of points from your comments and our responses.

Point 1: I identified that different mammary lineages representing the various molecular subtypes were used, however the number of satisfactory lineages in this study, which are four, I consider insufficient to cover all subtypes, so I understand that a larger number of mammary lineages should be used, as well as specifying each lineage.

Response 1: Information was added to the discussion describing the molecular subtypes of the cancer cell lines used in this study to satisfy the reviewer’s request to specify lineages of cell lines used. As the reviewer notes, the five cell lines used represent four molecular subtypes – luminal A, luminal B, HER2-enriched, and triple negative. While there is room for expansion into different lineages within each of these subtypes, particularly within the triple negative subtype, we believe that the breadth of molecular variability is sufficient for the present study. We look forward to expanding the breadth of lineages in future studies of SSA.

Point 2: The methodology described requires more information such as the number of animals used, informing tumor characteristics (subtype) after induction of the drug, N-methyl-N-nitrosourea (MNU).

Response 2: Additional information was added to the appropriate portions of the materials and methods, results, and discussion sections to better describe the animal study design and characteristics of the model.

Round 2

Reviewer 2 Report

I consider that the questions and suggestions requested have been answered. I am recommending to the editor, the publication.